# B-Box Transcription Factor FaBBX22 Promotes Light-Induced Anthocyanin Accumulation in Strawberry (*Fragaria* × *ananassa*)

**DOI:** 10.3390/ijms23147757

**Published:** 2022-07-14

**Authors:** Yongqiang Liu, Yuntian Ye, Yiping Wang, Leiyu Jiang, Maolan Yue, Li Tang, Mingsongxue Jin, Yunting Zhang, Yuanxiu Lin, Haoru Tang

**Affiliations:** 1College of Horticulture, Sichuan Agricultural University, Chengdu 611130, China; liuyq0129@163.com (Y.L.); yeyuntian@sicau.edu.cn (Y.Y.); wangyp_sicau@163.com (Y.W.); jiangleiyusicau@outlook.com (L.J.); maolanyue@outlook.com (M.Y.); tangli_sicau@163.com (L.T.); jmsx_sicau@163.com (M.J.); asyunting@sicau.edu.cn (Y.Z.); linyx@sicau.edu.cn (Y.L.); 2Institute of Pomology and Olericulture, Sichuan Agricultural University, Chengdu 611130, China

**Keywords:** strawberry, BBX22, HY5, anthocyanin accumulation, light

## Abstract

B-box transcription factors (TFs) play a vital role in light-induced anthocyanin accumulation. Here, the *FaBBX22* gene encoding 287 amino acids B-box TF was isolated from the cultivated strawberry variety ‘Benihoppe’ and characterized functionally. The expression analysis showed that *FaBBX22* was expressed in the roots, stems, leaves, flowers and fruits, and its transcription level was upregulated under the red- or blue-light irradiation. FaBBX22 was localized in the nucleus and showed trans-acting activity in yeast cells. Ectopic overexpression of *FaBBX22* in *Arabidopsis* enhanced the accumulation of anthocyanin. Additionally, we obtained transgenic strawberry calli that overexpressed the *FaBBX22* gene, and strawberry calli coloration assays showed that FaBBX22 increased anthocyanin accumulation by upregulating the expression of anthocyanin biosynthetic genes (*FaPAL*, *FaANS*, *FaF3′H*, *FaUFGT1*) and transport gene *FaRAP* in a light-dependent manner. Yeast two-hybrid (Y2H) and bimolecular fluorescence complementation assays indicated that FaBBX22 interacted with FaHY5. Furthermore, mutation of the 70th Asp residue in FaBBX22 protein to an Ala residue disrupted the interaction between FaBBX22 and FaHY5. Further, a transient expression assay demonstrated that the co-expression of FaBBX22 and FaHY5 could strongly promote anthocyanin accumulation in strawberry fruits. Collectively, these results revealed the positive regulatory role of FaBBX22 in light-induced anthocyanin accumulation.

## 1. Introduction

Cultivated strawberry (*Fragaria* × *ananassa*) is widely grown worldwide for its nutrients, unique flavor and economic value. The color of strawberry fruits acts as an important character of its quality, and is principally determined by the type and content of anthocyanins. Anthocyanins belong to flavonoid compounds, which play an important role in flower pollination, seed dispersal and resistance to environmental stresses [1]. Moreover, numerous studies have indicated that anthocyanins have potential health benefits, due to their antioxidant activity [2,3].

The biosynthesis of anthocyanins involves a series of enzymatic reactions. The structural genes that encode the anthocyanin biosynthetic enzymes have been identified in strawberry, including *PAL* (phenylalanine ammonia lyase), *CHS* (chalcone synthase), *CHI* (chalcone isomerase), *F3H* (flavanone-3-hydroxylase), *F3′H* (flavonoid 3′-hydroxylase), *DFR* (dihydroflavonol 4-reductase), *ANS* (anthocyanidin synthase) and *UFGT* (UDP-glucose flavonoid 3-O-glucosyltransferase) [4,5,6,7,8,9,10]. Recently, several glutathione S-transferases (GST) have been isolated in plants, which are responsible for the transport of anthocyanin from the cytosol to vacuole [11,12,13]. For example, *RAP* (reduced anthocyanins in petioles) encode a GST transporter in strawberry, and its mutation will prevent anthocyanins from accumulating in foliage and fruit [14,15]. Additionally, several regulatory genes, which control the expression of anthocyanin-related genes, are confirmed functionally in strawberry. Among them, MYB TFs (FaMYB10, FaGAMYB), bHLH TFs (FvbHLH33, FvbHLH9), bZIP TF (FvHY5), NAC TF (FaRIF), MADS-box TF (FaSHP), AP2/ERF TF (FaRAV1) and ABA-stress-ripening protein (FaASR) have a positive effect on anthocyanin accumulation [16,17,18,19,20,21,22,23]. The overexpressing or silencing of their expression will alter the content of the anthocyanins in strawberry. Moreover, FaMYB1, a R2R3-MYB TF, plays a negative role in regulating the expression of late anthocyanin biosynthetic structural genes [24]. However, the anthocyanin-transcriptional regulation network remains to be further improved in strawberry.

Besides the genetic components, the anthocyanin biosynthesis is also affected by environmental factors. Numerous studies have shown that light exposure can promote the accumulation of anthocyanin in most fruit crops, while shading does the opposite [25]. In strawberry, light is essential for anthocyanin accumulation, and light exposure treatment can upregulate the expression of the anthocyanin-related genes, such as *CHS*, *CHI*, *DFR*, *F3H*, *ANS*, *UFGT* and *MYB10* [26]. Additionally, the specific wavelengths of light have an important effect on anthocyanin accumulation, especially the short wavelength light. In apple, blue light and UV-B irradiation induce anthocyanin accumulation in the fruit peel, but red light has little effect [27]. Similar reports have been shown in pear, peach, grape and sweet cherry [28,29,30,31]. In our previous work, the total anthocyanins’ content (TAC), pelargonidin 3-glucoside (Pg3G) and pelargonidin 3-malonylglucoside (Pg3MG) of strawberry fruits was significantly increased by blue and red light-emitting diodes (LED) treatment [32]. Miao et al. (2016) treated strawberry plants with colored light-quality selective plastic films, and they found that the red and yellow films increased the TAC compared to the control treatment (white films), while the green and blue films led to a reduction of the TAC in fruits [33]. Further assays indicated that a higher anthocyanin accumulation was associated with higher enzymes’ activity and expression of anthocyanin-related genes. As a part of photomorphogenesis, light signals regulating the accumulation of anthocyanin has been widely studied in *Arabidopsis*, which includes light sensing and signal transduction. The photoreceptors are responsible for perceiving the different wavelengths of light, including phytochromes (far red/red light receptor), cryptochromes and phototropins (blue/UV-A receptor) and UVR8 (UV-B receptor) [34]. The light-activated photoreceptors initiate downstream light signal transduction through the protein–protein interactions with light-responsive elements. These light-responsive elements include the CONSTITUTIVE PHOTOMORPHOGENIC1 (COP1), SUPPRESSOR OF PHYA (SPA) and many TFs [34]. COP1, an E3 ubiquitin ligase, interacts with the light-responsive regulators of anthocyanin biosynthesis, such as ELONGATED HYPOCOTYL5 (HY5), MYB TFs and B-box TFs, and subsequently mediates the interaction partner ubiquitination and degradation by the 26S proteasome pathway in darkness [35,36,37].

The B-box (BBX) TFs belong to the zinc finger-proteins family containing one or two B-box motifs, and sometimes also feature a CCT (CONSTANS, CO-like and TOC1) domain [38]. The BBX proteins are involved in numerous biological processes that include seeding photomorphogenesis, flowering and the response to biotic and abiotic stresses [39]. In addition, a role in anthocyanin biosynthesis has been shown for BBX TFs in *Arabidopsis* and a few horticultural plants [40,41,42,43]. In most cases, the BBX proteins converge on light-responsive factor HY5 in the regulation of anthocyanin biosynthesis [44]. In *Arabidopsis*, HY5 is a central modulator of light signaling and regulates light-induced anthocyanin-related genes, such as *MYB75*, *CHS*, *CHI* and *F3H* [45]. Chang et al., 2008, reported that HY5 has a positive trans-acting activity on *AtBBX22*, an activator of light-induced anthocyanin accumulation, and binding affinity to its promoter [41]. In addition, the AtBBX21 and HY5 directly bind to the T/G-box cis-element present in the HY5 promoter to activate *HY5* expression, and so promote anthocyanin biosynthesis under the light [46]. On the other hand, both AtBBX21 and AtBBX22 interact with HY5, which enhances the induction effect of HY5 on the downstream target genes [44]. In contrast, several BBX proteins interact with HY5 and hamper the binding of HY5 to its target genes, thereby repressing anthocyanin biosynthesis, such as AtBBX24 in *Arabidopsis*, MdBBX37 in apple and PpBBX21 in pear [47,48,49]. Meanwhile, the BBX proteins can independently regulate the expression of anthocyanin structural genes by directly binding to their promoter [50,51].

In our previous work, a total of 51 members of the BBX family were identified in octoploid strawberry, and a BBX gene (*FaBBX22*) was found to be differentially expressed during light quality treatment on strawberry [52]. In the present study, the *FaBBX22* gene was isolated from the cultivated strawberry variety ‘Benihoppe’. We analyzed the expression pattern, subcellular localization and transcriptional activity of FaBBX22. Next, the heterologous expression of *FaBBX22* in *Arabidopsis* increased the content of anthocyanins in leaves and floral stems. Moreover, the overexpression of *FaBBX22* in the strawberry calli indicated that FaBBX22 promoted anthocyanin accumulation in a light-dependent manner. Furthermore, the protein interaction between FaBBX22 and FaHY5 was verified. A transient expression assay in white strawberry showed that *FaBBX22* strongly promoted anthocyanin accumulation in fruit when co-expressed with *FaHY5*. These results confirmed the role of FaBBX22 in strawberry anthocyanin accumulation and enriched our understanding of the regulatory network of anthocyanin metabolism.

## 2. Results

### 2.1. Isolation and Sequence Analysis of FaBBX22

In our previous study, a total of 51 members of the BBX family were identified in octoploid strawberry [52], but most of their functions are unknown. In this work, specific primers were designed to amplify the *FaBBX22* gene, according to the transcriptome data described in the previous study [53]. The CDS of the *FaBBX22* gene was obtained from the cultivated strawberry variety ‘Benihoppe’ (Appendix A). FaBBX22 contained an open reading frame (ORF) of 864 bp encoding 287 amino acids and the predicted protein had a molecular mass of 30.9 kDa with a theoretical isoelectric point (pI) of 5.12. The phylogenetic analysis showed that FaBBX22 was clustered with RcBBX22, MdBBX22, PavBBX22 and PpBBX22 in rosaceous plants, suggesting that BBX22 are relatively conserved evolutionarily (Figure 1A). The multiple sequence alignment of the amino acids showed that FaBBX22 harbored conserved B-box domains at its N-terminal (Figure 1B). The three conserved aspartic acid (Asp) residues, which are important for B-box proteins interaction with other proteins, were found in FaBBX22 (Figure 1B).

### 2.2. Expression Pattern of FaBBX22

The spatial and temporal expression patterns of *FaBBX22* were analyzed by qRT-PCR (Figure 2A). Our results showed that *FaBBX22* could be detected in the roots, stems, leaves, flowers and fruits of the ‘Benihoppe’ strawberry. *FaBBX22* has the highest expression level in the roots and a relatively lower transcript abundance in the stems and fruits. During the fruits’ development, the expression level of *FaBBX22* in the fruit coloring stage was slightly higher than in the green and white fruit stages.

The expression patterns of *FaBBX22* during the red and blue light irradiation treatments were further analyzed (Figure 2B). In our experiment, the expression of *FaBBX22* was immediately upregulated (within 6 h) in both the red and blue light exposure. After 6 h, *FaBBX22*’s expression gradually decreased under the red light exposure, while the expression pattern of FaBBX22 was down-up-down and reached its peak at 24 h under the blue light exposure. These results demonstrated that *FaBBX22* is a light-responsive gene and its expression pattern is different under various light quality irradiations.

### 2.3. Subcellular Localization and Transcriptional Activity Analysis of FaBBX22

To test the subcellular localization of FaBBX22, a transient co-transformation of strawberry protoplasts with the d35S::FaBBX22-EGFP (pYTSL-16-FaBBX22) and a d35S::NLS-RFP (nuclear location marker) construct was performed. Figure 3A shows that a clear green fluorescence signal was only detected in the nucleus.

A yeast assay was performed to determine the transcriptional activity of FaBBX22 (Figure 3B). The yeast strain Y2HGold transformed with pGBKT7-FaBBX22 (validation group) or pGBKT7-Lam (a negative control) construct grew well on the SD/-Trp medium, but only the validation group was able to grow on the SD/-Trp-His-Ade medium and turned blue with X-α-Gal. These results indicated that FaBBX22 had trans-acting activity in yeast cells.

### 2.4. Ectopic Expression of FaBBX22 in Arabidopsis

To verify the functions of FaBBX22, the 35S::FaBBX22-Flag construct was transformed into *Arabidopsis thaliana* (Col-0) and two independent transgenic lines were obtained (Figure 4). The expression levels of *FaBBX22* in transgenic lines were detected by semi-quantitative RT-PCR, which indicated that *FaBBX22* was successfully overexpressed (Figure 4B). Compared to the wild type, the transgenic lines showed a higher anthocyanin content in the leaves and floral stems under the light (Figure 4C). Furthermore, the expression levels of the anthocyanin biosynthetic structural genes were upregulated, which was consistent with the phenotype of the transgenic lines (Figure 4D). These results indicated that FaBBX22 promoted anthocyanin accumulation in *Arabidopsis*.

### 2.5. Overexpression of FaBBX22 in Strawberry Calli

To further confirm the role of FaBBX22 in anthocyanin accumulation of strawberry, we overexpressed *FaBBX22* in the ‘Benihoppe’ strawberry and obtained the transgenic calli lines (*FaBBX22*-OX) (Figure 5). As Figure 5A shows, the *FaBBX22*-OX and empty vector control (EV) calli were a light yellow color under the dark. However, the *FaBBX22*-OX calli turned into a red color, which was cultured in the continuous white light for 7 days, but not in the empty vector calli (Figure 5A). As observed, the anthocyanin content was significantly higher in the *FaBBX22*-OX calli under the light conditions than the others (Figure 5C). We also analyzed the expression levels of the *FaBBX22* and anthocyanin-related genes by qRT-PCR (Figure 5B,D). As expected, the *FaBBX22* was highly expressed under both the dark and light conditions in the *FaBBX22*-OX calli, but some of the anthocyanin biosynthetic structural genes were only highly expressed under the light conditions, such as *FaPAL*, *FaANS*, *FaF3′H* and *FaUFGT1*. In addition, we found that *FaRAP*, an anthocyanin transport gene in strawberry, was upregulated by *FaBBX22*’s overexpression in a light-dependent manner. Taken together, these results indicated that FaBBX22 was involved in the light-induced anthocyanin accumulation in strawberry.

### 2.6. FaBBX22 Interacts with FaHY5

Some of the BBX proteins, usually as HY5-associated proteins, participate in the regulation of anthocyanin accumulation through the BBX-HY5 heterodimer [54]. Thus, a yeast two-hybrid assay was performed to investigate whether FaBBX22 can interact with HY5 (Figure 6). Our results showed that FaBBX22 had trans-acting activity in the yeast strain Y2HGold (Figure 3B), so the FaBBX22 coding sequence was ligated to the pGADT7 vector. The full-length coding sequence of FaHY5 was cloned and inserted to the pGBKT7 vector for transcriptional activity assays. As has been reported in other plants [49,55], FaHY5 did not possess trans-acting activity in yeast cells. Then, the pGADT7-FaBBX22 and pGBKT7-FaHY5 constructs were co-transformed into Y2HGold, which grew well on the SD/-Trp-Leu-His-Ade medium with 100 ng/mL Aureobasidin A and turned blue with X-α-Gal. This result indicated that FaBBX22 interacted with FaHY5, which was confirmed by a bimolecular fluorescence complementation assay in strawberry protoplasts (Figure 6C).

The Asp residues in the B-box domains of the BBX proteins play an important role in the physical interaction with HY5 proteins [49,56]. As shown in Figure 6A, we mutated the three Asp residues in the B-box domains of FaBBX22 to alanine (Ala) residues, and designated these proteins FaBBX22-D20A, FaBBX22-D70A and FaBBX22-D79A, respectively. Further yeast two-hybrid assays showed that the mutation of the 20th or 79th in FaBBX22 protein to Ala residue did not affect the interaction between FaBBX22 and FaHY5, while the mutation of the 70th Asp residue to Ala did (Figure 6B). These results indicated that the 70th Asp residues of FaBBX22 were important for the formation of the FaBBX22-FaHY5 heterodimer.

### 2.7. Overexpression of FaBBX22 and FaHY5 in Strawberry Receptacles

To investigate the effects on anthocyanin accumulation of the physical interaction between FaBBX22 and FaHY5, a transient gene expression system was performed in strawberry receptacles (Figure 7). We chose a white fruit strawberry variety, ‘Snow white’, whose fresh fruit are completely white in color, to observe the accumulation of the pigments more clearly. In our study, the transient overexpression of *FaBBX22* (*FaBBX22*-OX) or *FaHY5* (*FaHY5*-OX) alone could not significantly promote anthocyanin accumulation (Figure 7A). However, when the *Agrobacteria* harboring *FaBBX22* were co-infiltrated with the *Agrobacteria* harboring *FaHY5*, there was a noticeable redness in the white fruit flesh and the total anthocyanins content was significantly higher than that of a single or empty vector infiltration (Figure 7A,B). Further, we evaluated the expression levels of *FaHY5*, *FaBBX22* and three anthocyanin-related genes by qRT-PCR (Figure 7C,D). As expected, *FaHY5* and *FaBBX22* were successfully overexpressed in the strawberry receptacles, respectively. Compared with the empty vector control, the *FaHY5* was upregulated 3.1-fold in the receptacles’ overexpression in *FaBBX22* alone. Meanwhile, two of the key structural genes of anthocyanin biosynthesis (*FaCHS* and *FaUFGT1*) and the anthocyanin transporter *FaRAP* also had an upregulation greater than the control in receptacles’ overexpression of *FaBBX22* alone. However, the overexpression of *FaHY5* alone only significantly upregulated the *FaUFGT1* gene. The co-infiltration of *FaBBX22* with *FaHY5* significantly enhanced the expression levels of *FaCHS*, *FaUFGT1* and *FaRAP*, which was consistent with the anthocyanin accumulation in the strawberry receptacles.

## 3. Discussion

Anthocyanins are one of the important substances affecting the quality of horticultural products, and they are also widely involved in the process of plants’ resistance to stresses as an antioxidant. Many transcription factors have been reported to be involved in the anthocyanin metabolic pathway in strawberry, including MYB, bHLH, bZIP, MADS-box, etc., but no BBX transcription factors have been found [16,17,18,19,20,21,22,23]. In this study, FaBBX22, a light-responsive BBX TF, was cloned from the cultivated strawberry ‘Benihoppe’. The anthocyanins’ content of the transgenic lines that overexpressed *FaBBX22* was significantly higher than control. This is the first report that the BBX gene regulates anthocyanin biosynthesis in strawberry, which enriches our understanding of the anthocyanin biosynthesis at the transcriptional level.

In *Arabidopsis*, Gangappa et al. (2014) identified 32 BBX family members, of which AtBBX20-25 and AtBBX32 were shown to regulate anthocyanin biosynthesis [40,44,47,56,57,58,59]. The amino acid sequence alignment between FaBBX22 and AtBBXs showed that FaBBX22 and AtBBX22 have the highest sequence similarity. AtBBX22, also known as light-regulated zinc finger 1 (LZF1)/salt tolerance homolog 3 (STH3), has been reported to positively affect anthocyanin biosynthesis [41,56]. Furthermore, PpBBX16, a homolog of AtBBX22 in red pear, was also identified as a positive regulator of light-induced anthocyanin accumulation [43]. In this study, a heterologous transgenic study was conducted by overexpressing *FaBBX22* in *Arabidopsis*. In contrast to the wild type, the transgenic leaves and floral stems of *FaBBX22*-OX were purple with more anthocyanins. Similarly, apple MdCOL11 is a homolog of *Arabidopsis* AtBBX22, and its overexpression enhanced the accumulation of anthocyanin in *Arabidopsis* [42]. Therefore, the above results indicate that the *BBX22* genes have a conserved function in anthocyanin metabolism. Additionally, we found that FaBBX22 was a nuclear-localized protein and had trans-acting activity in yeast cells. Thus, FaBBX22 may function as a transcription activator and regulate the anthocyanin-related genes. Indeed, FaBBX22 activated the anthocyanin biosynthetic structural genes in transgenic *Arabidopsis* lines, including *AtCHS*, *AtCHI*, *AtF3H*, *AtDFR* and *AtLDOX*, that is consistent with the BBX proteins in *Arabidopsis,* which can increase the anthocyanin content by regulating the expression levels of the structural genes [49]. Meanwhile, we also obtained transgenic strawberry calli overexpressing *FaBBX22* under the control of the cauliflower mosaic virus 35S promoter. As hypothesized, the anthocyanin content in the strawberry calli overexpressed with *FaBBX22* was higher than that in the empty control, but light was required. Further qRT-PCR analysis indicated that the expression levels of several structural genes, such as *FaPAL*, *FaANS*, *FaF3′H* and *FaUFGT1*, were clearly upregulated by *FaBBX22*’s overexpression in a light-dependent manner. In strawberry, the *FaRAP* gene encodes a GST protein and is responsible for anthocyanin transport from cytosol to vacuole [14,15]. Interestingly, the expression level of *FaRAP* was also upregulated under light conditions by *FaBBX22*’s overexpression. All of these results suggested that FaBBX22 may participate in the regulation of anthocyanin accumulation in a light-dependent manner.

Numerous studies have shown that light intensity, light quality and the photoperiod are able to affect anthocyanin biosynthesis [25]. In most fruit trees, shading treatment reduces the content of anthocyanins, especially in the fruit skin. For instance, mature woodland strawberry fruits could not accumulate anthocyanin under dark conditions, and the expression levels of *FvCHS*, *FvF3H*, *FvCHI*, *FvDFR* and *FvLDOX* were significantly down-regulated [26]. The light quality has different effects on anthocyanin biosynthesis. The shorter wavelength light is more likely to cause anthocyanin accumulation in fruits. Different cultivars of peach have differential sensitivity to light. ‘Hujingmilu’, a naturally deeply colored cultivar, can be induced to biosynthesize anthocyanin by UV-A and UV-B, while ‘Yulu’, a low pigmentation peach, is sensitive to UV-B only, but not to UV-A [29]. Postharvest treatments with light irradiation of different wavelengths on strawberry have been performed to increase the anthocyanin content, and blue light (465 nm) irradiation had a better effect than red (660 nm) and green light (535 nm) [60]. In our previous study, we also found that the red and blue light significantly increased the total anthocyanin content of strawberry fruits compared to white light [32]. Furthermore, high performance liquid chromatography (HPLC) analysis indicated that the increased total anthocyanins mainly depended on pelargonidin 3-glucoside production [32]. In the present study, the expression of *FaBBX22* was induced by the blue and red light irradiation treatments, and its expression trend under the blue light treatment was higher than the red light treatment. Therefore, we hypothesized that FaBBX22 may be involved in the light-quality regulation of anthocyanin accumulation in strawberry. Additionally, *FaBBX22* was differentially expressed in different tissues of strawberry, and the distinct functions of *FaBBX22* in the different tissues and organs need to be further explored, based on the phenotypes of transgenic strawberry plants.

Sometimes the BBX proteins require the assistance of a partner protein, such as HY5, to regulate the expression of target genes [39]. HY5, a key regulator of light signal transduction in plants, is generally thought to act as a DNA-binding transcriptional regulator, due to its lack of any apparent transactivation domain [61]. In *Arabidopsis*, Job et al. (2018) showed that AtBBX21 interacts with AtHY5, and the two proteins can coordinate to promote photomorphogenesis, including anthocyanin accumulation [47]. Recently, the *bbx20 bbx21 bbx22* triple mutant was obtained, which has phenotypes that are similar to *hy5* mutant, and 84% of genes that exhibit differential expression in *bbx20 bbx21 bbx22* are also regulated by HY5 [44]. In pear, PpBBX16 could not directly induce the expression of anthocyanin-related genes by itself, but needed PpHY5 to gain full function [43]. In the present study, we demonstrated the interaction between the FaBBX22 and FaHY5 protein, using yeast two-hybrid and bimolecular fluorescence complementation. The B-box domains are important for the BBX proteins’ interaction with other proteins. Datta et al. (2008) mutated the three individual Asp residues in the B-box domains of AtBBX22 to Ala, and they found that the Asp residues in the B-box2 motif are required for AtBBX22 to interact with AtHY5 [56]. The amino acid sequence alignment of FaBBX22 with other known BBXs showed that the B-box domains of these proteins were highly similar. Similar to AtBBX22, we also identified three conserved Asp residues (located in the 20th, 70th and 79th positions of the protein) in the B-box domain of FaBBX22. Further analysis revealed that the mutation of the 20th or 79th Asp residue in FaBBX22 protein to Ala did not affect the interaction between FaBBX22 and FaHY5, while the mutation of the 70th Asp residue to Ala did. Therefore, we speculated that the substitution of the key amino acid residues is likely to disrupt the structure of the B-box, thus affecting the interaction relationship. To further clarify the interaction of FaBBX22 and FaHY5, they were transiently overexpressed in receptacles of the ‘Snow white’ strawberry. The transient expression of *FaBBX22* or *FaHY5* alone could not induce anthocyanin accumulation, whereas the co-expression of *FaBBX22* and *FaHY5* strongly promoted anthocyanin accumulation in white-flesh fruits. These results suggested that FaHY5 plays a potentially important role in the regulation of anthocyanin accumulation by FaBBX22. PpHY5 can directly bind to the promoter of *PpMYB10*, a key MYB transcription factor for the red coloration of pear fruits, but which requires the BBX proteins for transcriptional regulation and, subsequently, regulates the anthocyanin biosynthesis [43]. In apple, a model of BBXs-HY5 heterodimers was also established, for regulating the anthocyanin accumulation by affecting the expression of *MdMYB1*, a homolog of *PpMYB10*, [50,54,62]. Similar to other rosaceous fruits, several studies have shown that MYB10 is the main activator for anthocyanin biosynthesis in strawberry fruits [63]. The completely white flesh of the ‘Snow white’ strawberry is caused by the loss of function of *FaMYB10* [64]. In the present work, the co-infiltration of *FaBBX22* and *FaHY5* led to a recovery of anthocyanin accumulation in the ‘Snow white’ fruits. Thus, our results suggested that the FaBBX22-FaHY5 heterodimer may be independent of FaMYB10 in the regulation of anthocyanin accumulation, but the detailed mechanisms need to be further explored.

## 4. Materials and Methods

### 4.1. Plant Materials and Light Treatment

The strawberry (*Fragaria* × *ananassa*) variety ‘Benihoppe’ was grown in a simple greenhouse of the Sichuan Agricultural University, Chengdu, China. The roots, stems, leaves, flowers and fruits were collected to analyze tissue-specific gene expression. According to the maturity of the strawberry fruit, five stages of fruit (GF, green fruit; WF, white fruit; TR, turning red; HR, half red; FR, full red) were harvested at 15, 22, 28, 34 and 40 days after anthesis.

The strawberry fruits were harvested as uniform size, no mechanical damage with the sepals and stamens removed for light treatment. All of the fruits were placed in an artificial climate chamber (10 °C, relative humidity 70%) to balance in the dark. After 2 days, two/three fruits were treated with red and blue light and the remaining fruits were still in the dark. The samples were taken at 0, 6, 12, 18, 24 and 48 h, respectively. Six individual fruits were used for each time point and independently replicated three times. All of the samples were frozen with liquid nitrogen and stored at −80 °C.

### 4.2. Gene Isolation and Sequence Analysis

The total RNA of the strawberry tissues was extracted by a modified CTAB method [65]. The first strand of cDNA was synthesized by using RT Easy^TM^ II with gDNase Reagent Kit (Foregene, Chengdu, China). The gene sequence of *FaBBX22* was obtained according to the transcriptome data of the ‘Benihoppe’ strawberry. Cloning primers were designed by SnapGene software, as shown in Appendix A. The full-length coding sequence of FaBBX22 was amplified by PCR. Then, the PCR products were cloned into the *pEASY*-Blunt Cloning vector (TransGen, Beijing, China) and sequenced (Sangon, Chengdu, China). The amino acid sequence of the FaBBX22’s orthologues from other plant species were obtained by BLASTP in NCBI (https://www.ncbi.nlm.nih.gov/) (accessed on 23 December 2021). MEGA 7.0 software was used to construct the phylogenetic tree [66]. Multiple sequences of FaBBX22 and its homologous proteins were compared, using DNAman 7.0 software (Lynnon, QC, Canada).

### 4.3. Strawberry Protoplasts Isolation and Transient Transformation

The strawberry protoplasts were isolated by the enzymes digestion method. The enzyme solution was 11% CPW (cell protoplast wash medium) buffer containing 0.05% MES (*w*/*v*), 2% Cellulose R-10 (*w*/*v*), 1% Macerozyme R-10 (*w*/*v*) and 0.32% Pectolyase Y-23 (*w*/*v*). The 0.5–1 mm leaf strips were cut from the well-expanded leaves of ‘Benihoppe’ strawberry (~40-day-old) using a fresh sharp surgical blade. Then, the leaf strips were removed quickly and gently into the enzyme solution (10 leaves in 5 mL), and incubated in the dark at 28 °C for 12 h. After digestion, the enzyme solution should turn green, which indicates the release of the protoplasts. The strawberry protoplasts’ purification and transformation were performed following Yoo et al. (2007) for *Arabidopsis* [67].

### 4.4. Subcellular Localization

The full-length CDS of *FaBBX22* was inserted to plant expression vector pYTSL-16, which is an enhanced green fluorescent protein (EGFP) fusion construct driven by the double CaMV 35S promoter. To verify the subcellular localization of FaBBX22, the strawberry protoplasts were co-transformed with pYTSL-16-FaBBX22 (d35S::FaBBX22-EGFP) and a nuclear location marker (d35S::NLS-RFP). After 18 h, the strawberry protoplasts were collected and visualized, using a confocal laser scanning microscope (FV3000, OLYMPUS, Tokyo, Japan).

### 4.5. Transcriptional Activity Assay

The full-length CDS of *FaBBX22* was inserted to yeast expression vector pGBKT7 (BD). To verify the transcriptional activity of FaBBX22, the pGBKT7-FaBBX22 vector and negative control (pGBKT7-lam) were transformed to yeast strain Y2HGold, respectively. After 3 days, the growth of the yeast cells in the synthetically defined medium (SD/-Trp and SD/-Trp-Ade-His/X-α-gal) was observed.

### 4.6. Transformation of Arabidopsis

For the overexpression of *FaBBX22*, the full-length CDS of *FaBBX22* was inserted to the modified pCAMBIA1301-3×Flag plant expression vector that was driven by the CaMV 35S promoter. This recombinant vector pCAMBIA1301-FaBBX22-Flag (35S::FaBBX22-Flag) was stably transformed into *Arabidopsis thaliana* (Col-0) via an *Agrobacterium*-mediated floral dip method [68]. The transgenic lines overexpressing the FaBBX22-Flag were screened on Murashige–Skoog (MS) medium containing 50 mg/L hygromycin. T3 plants were used for characterization.

### 4.7. Generation of Transgenic Strawberry Calli

The ‘Benihoppe’ strawberry was selected for the calli transformation. The young strawberry leaves were drilled with a 4 mm diameter punch, and then placed in pre-cultivation medium (1 × MS, 2% sucrose, 4 mg/L thidiazuron (TDZ), 0.5 mg/L indole-3-butyric acid (IBA) and 0.7% agar, pH 5.8) for 3 days in the dark. The construct pCAMBIA1301-FaBBX22-Flag was transformed into *Agrobacterium tumefaciens* GV3101. The strawberry leaf discs were immersed into *Agrobacterium* (OD_600_ = 0.5) for 20 min in co-cultivation buffer (1 × MS, 2% sucrose, pH 5.6), and then transferred to the co-cultivation medium (pre-medium supplemented with 100 μmol/L acetosyringone), and kept in the dark. After 3 days, the strawberry leaf discs were transferred to a regeneration medium (pre-medium supplemented with 250 mg/L carbenicillin and 200 mg/L timentin), and kept in the dark for 7 days. Next, the strawberry leaf discs were transferred to new selective medium (regeneration medium supplemented with 5 mg/L hygromycin) every 20 days until a large number of transgenic strawberry calli appeared. For the light treatment, the newly sub-cultured empty vector and the FaBBX22-containing calli were moved to the continuous white light conditions for 7 days and then used for observation.

### 4.8. Yeast Two-Hybrid Assay

The yeast two-hybrid assays were performed, according to the manufacturer’s instructions, using the Matchmaker Gold Yeast Two-Hybrid System kit (Takara, Beijing, China). Briefly, FaBBX22 and its protein mutants were fused to the active domain of GAL4 (AD), and FaHY5 were fused to the DNA-binding domain of GAL4 (BD). The BD and AD plasmids were co-transformed into yeast strain Y2HGold, and, respectively, coated in synthetically defined medium (SD/-Trp-Leu and SD/-Trp-Leu-Ade-His/AbA/X-α-gal) for observation.

The point mutation constructs, FaBBX22-D20A, FaBBX22-D70A and FaBBX22-D79A, were generated with ClonExpress MultiS One Step Cloning Kit (Vazyme, Nanjing, China). The fragments of FaBBX22 containing D20A, D70A or D79A mutations were amplified using the primer pairs D20A-1/2, D70A-1/2 and D79A-1/2, respectively. After the PCR amplification, the products were ligated into pGADT7 vector, based on homologous recombination. The primer sequences used for the vector construction are listed in Appendix A.

### 4.9. BIFC Assay

For the bimolecular fluorescence complementation assay, the coding sequence of *FaBBX22* without termination codon was cloned into the nYFP (pXY103) vector, and the coding sequence of FaHY5 without termination codon was cloned into cYFP (pXY104) vector. Then, the constructs were transiently co-expressed into the strawberry protoplasts. The fluorescent signals were observed using a confocal laser scanning microscope (FV3000, OLYMPUS).

### 4.10. Transient Transformation Analysis in Strawberry Fruit

‘Snow white’, a white fruit octoploid strawberry variety, was selected for transient transfection. The transfection of the strawberry fruit was conducted according to a previous report, with minor modifications [5]. *Agrobacterium tumefaciens* strain GV3101 containing *FaBBX22* and *FaHY5* construct was transiently expressed in detached fruits of strawberry (~3 weeks after pollination). After the transfection, the cut ends of the detached fruits were wrapped in moist absorbent cotton to prevent dehydration. Twenty individual fruits were used for each construct and independently replicated three times. Seven days after transfection, all of the fruits were observed and collected.

### 4.11. qRT-PCR Analysis

The expression level of the *FaBBX22* and anthocyanin-related genes were detected by real-time quantitative PCR (qRT-PCR) technology. The reaction mixture was comprised of 2 μL cDNA (5 ng/μL), 1 μL forward/reverse primer (10 μM), 5 μL 2 × Real PCR EasyTM SYBR Mix (Foregene, Chengdu, China) and 1 μL ddH_2_O. The qRT-PCR was conducted on a CFX96 touch real-time PCR detection system (Bio-rad, Hercules, CA, USA). The relative expression levels of the genes were calculated using the 2^−ΔΔCT^ method [69]. The *Actin* gene (accession: AB116565.1) and *AtPP2A* were used as the reference genes in strawberry and *Arabidopsis*, respectively. All of the qRT-PCR primer sequences are listed in Appendix A.

### 4.12. Anthocyanin Measurement

The anthocyanin measurement was conducted according to a previous report [14]. Approximately 0.5 g fresh tissue was ground in liquid nitrogen, and added to 5 mL extraction solution (methanol: H_2_O: formic acid: trifluoroacetic acid, 70:27:2:1), and kept at 4 °C in the dark. After 12 h, the supernatant was collected by centrifugation at 13,000 rpm for 10 min at 4 °C. Then, the absorbance was measured at 530 nm and 657 nm by UV spectrophotometer (UV-1800PC, MAPADA, Shanghai, China). The formula for the calculation of the anthocyanins content is as follows: Total anthocyanins = [A530 − (0.25 × A657)]/M, where A530 and A657 are the absorbance at the indicated wavelengths, and M is the fresh tissue weight (g).

### 4.13. Statistical Analysis

All of the experimental data were expressed as the mean ± standard deviations from the mean (SD). The statistical analysis was performed using Student’s *t*-test (** *p* < 0.01) and one-way analysis of variance (ANOVA) of IBM SPSS Statistics software, version 27.0 (IBM, Chicago, IL, USA).

## 5. Conclusions

In summary, we found a strawberry B-box transcription factor, FaBBX22, which was induced by light irradiation treatment at the transcription level. The ectopic overexpression of *FaBBX22* promoted the leaves and floral stems’ coloration and the anthocyanin biosynthesis in *Arabidopsis*. Furthermore, we obtained transgenic strawberry calli that overexpressed *FaBBX22*, and *FaBBX22* promoted anthocyanin accumulation by upregulating the expression of anthocyanin biosynthetic genes and transport genes in a light-dependent manner. Finally, we confirmed that the FaBBX22-promoted anthocyanin accumulation was dependent on the FaBBX22-FaHY5 heterodimer in white-flesh strawberry. The present results provide insight into the transcriptional regulation of FaBBX22 and light-induced anthocyanin accumulation in strawberry.

## Figures and Tables

**Figure 1 ijms-23-07757-f001:**
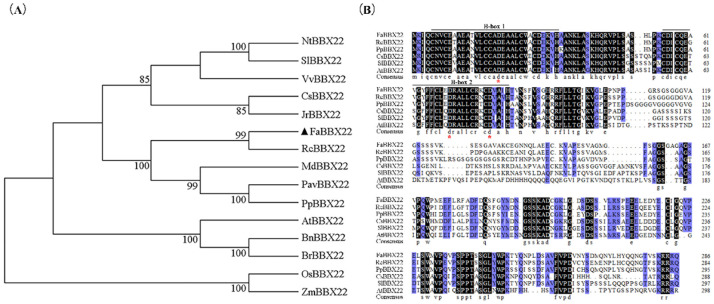
Alignment and phylogenetic tree of FaBBX22. (**A**) The phylogenetic relationships between the BBX22 proteins. The black triangle indicates FaBBX22. NtBBX22 (XP_016460367.1) in *Nicotiana tabacum*; SlBBX22 (XP_004244294.1) in *Solanum lycopersicum*; VvBBX22 (XP_002283666.1) in *Vitis vinifera*; CsBBX22 (XP_006477825.1) in *Citrus sinensis;* JrBBX22 (XP_018848786.1) in *Juglans regia;* RcBBX22 (XP_024157894.1) in *Rosa chinensis*; MdBBX22 (XP_028944947.1) in *Malus* × *domestica;* PavBBX22 (XP_021809667.1) in *Prunus avium;* PpBBX22 (XP_007211715.1) in *Prunus persica;* AtBBX22 (NP_565183.1) in *Arabidopsis thaliana;* BnBBX22 (XP_013649965.1) in *Brassica napus*; BrBBX22 (XP_009106574.1) in *Brassica rapa;* OsBBX22 (XP_015644333.1) in *Oryza sativa;* ZmBBX22 (XP_020395809.1) in *Zea mays*; (**B**) Multiple sequence alignment of amino acid sequence of FaBBX22 with other B-box TFs. The black line indicates B-box1 and B-box2 motifs, respectively. The red asterisk indicates aspartic acid residues in the B-box domain which are important for B-box TFs interaction with other proteins.

**Figure 2 ijms-23-07757-f002:**
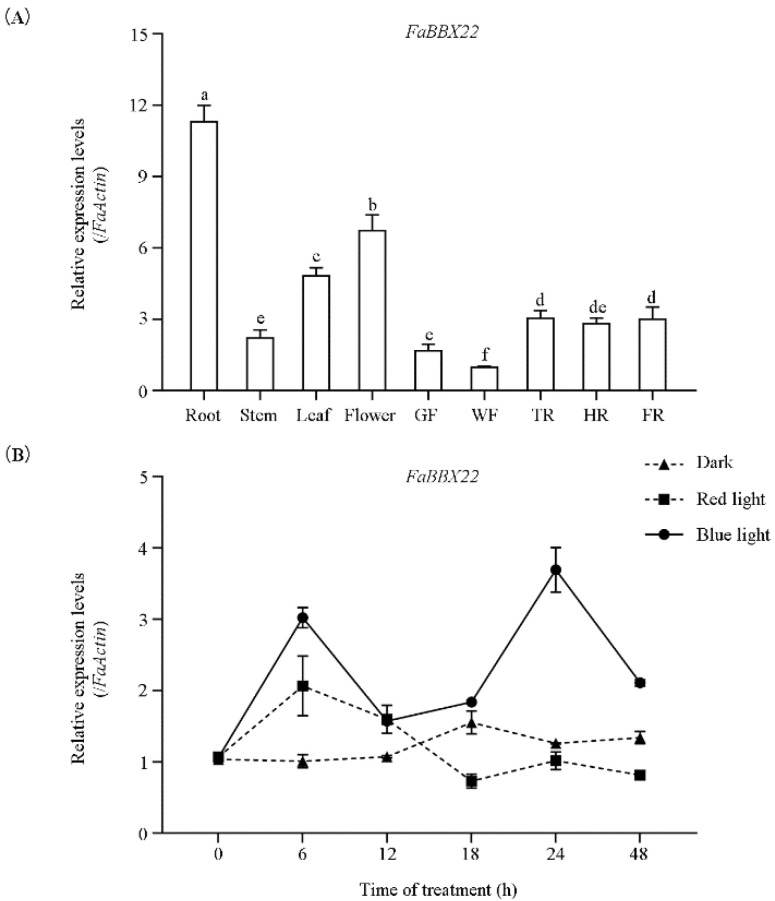
The expression pattern of *FaBBX22*. (**A**) The relative expression levels of *FaBBX22* in different tissues and fruit development stages. GF, green fruit; WF, white fruit; TR, turn red; HR, half fruit; FR, full fruit; (**B**) The relative expression levels of *FaBBX22* during red and blue light irradiation treatments. Error bars represent three independent replicates. Different letters above the bars indicate significantly different values (*p* < 0.05) according to a Least Significant Difference (LSD) test.

**Figure 3 ijms-23-07757-f003:**
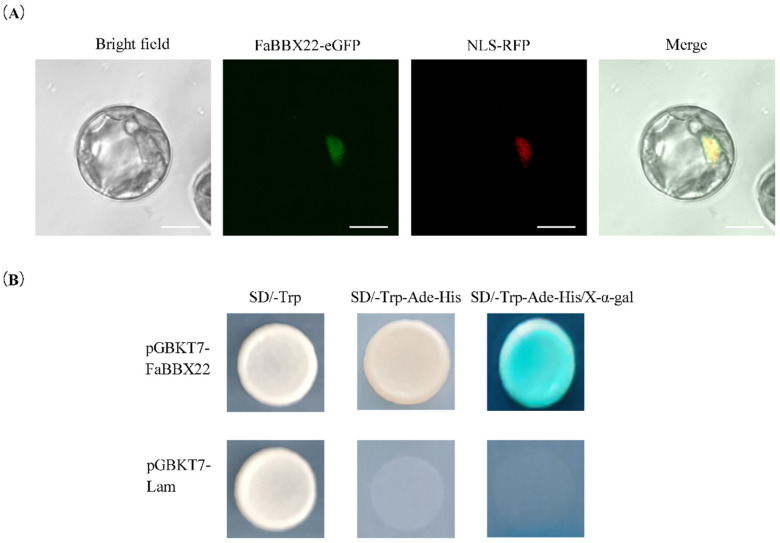
Transcription factor characteristics analysis of FaBBX22. (**A**) Subcellular localization of the FaBBX22 protein. Strawberry protoplasts were transformed with the fusion construct (FaBBX22-eGFP) and a nuclear marker NLS-RFP (NLS, nuclear localization sequence). Bars, 10 μm; (**B**) Transcriptional activity analysis of FaBBX22 in yeast cell. The yeast cells transformed with pGBKT7-Lam vector were used as a negative control.

**Figure 4 ijms-23-07757-f004:**
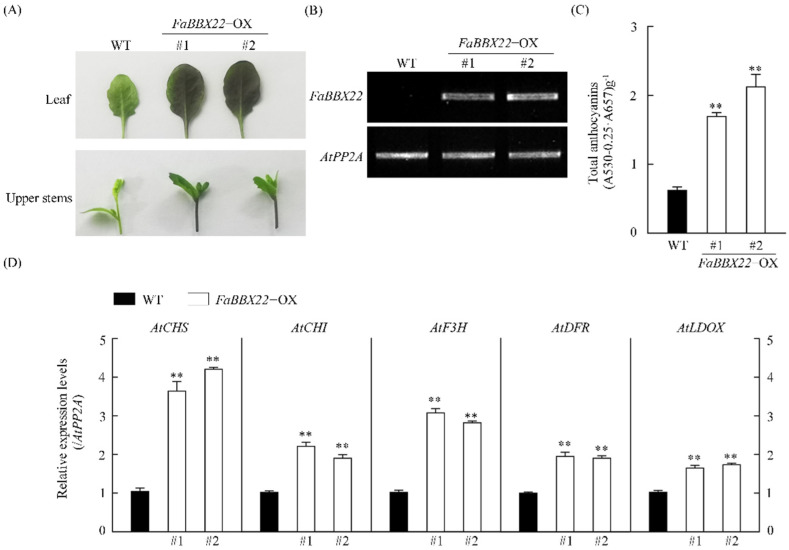
Effects of *FaBBX22*’s ectopic expression in *Arabidopsis*. (**A**) The phenotype comparison of wild−type (Columbia−0) and transgenic *Arabidopsis*. *FaBBX22*−OX represent *FaBBX22* overexpression lines. Two independent transgenic lines were used in the experiment; (**B**) The relative expression levels of *FaBBX22* by semi−qRT−PCR; (**C**) Anthocyanins content, and (**D**) expression levels of anthocyanin biosynthetic structural genes in wild-type and transgenic lines. Error bars represent the SD of three independent replicates. Asterisks indicate significant differences (two−tailed Student’s *t*−test, ** *p* < 0.01) compared with the wild-type control.

**Figure 5 ijms-23-07757-f005:**
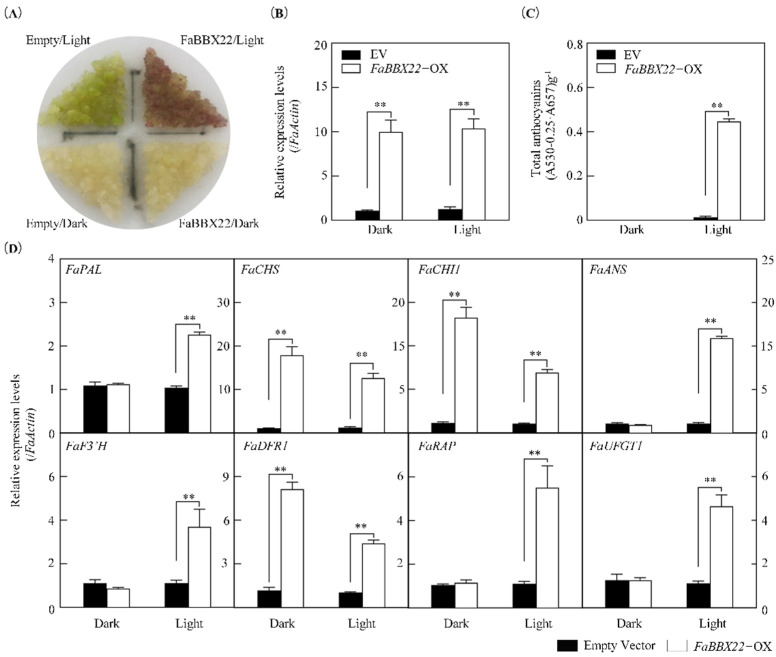
The effects of *FaBBX22* overexpression in strawberry calli. (**A**) *FaBBX22*’s overexpression resulted in an anthocyanin accumulation after 7 days of the light treatment; (**B**) *FaBBX22*’s expression level in transgenic strawberry calli; (**C**) Anthocyanins content in transgenic strawberry calli after the light treatment; (**D**) The expression levels of anthocyanin-related genes (*FaPAL*, *FaCHS*, *FaCHI1*, *FaANS*, *FaF3′H*, *FaDFR1*, *FaRAP*, *FaUFGT1*) in strawberry callis. Error bars represent the SD of three independent replicates. Asterisks indicate significant differences (two−tailed Student’s *t*−test, ** *p* < 0.01) compare with empty vector control.

**Figure 6 ijms-23-07757-f006:**
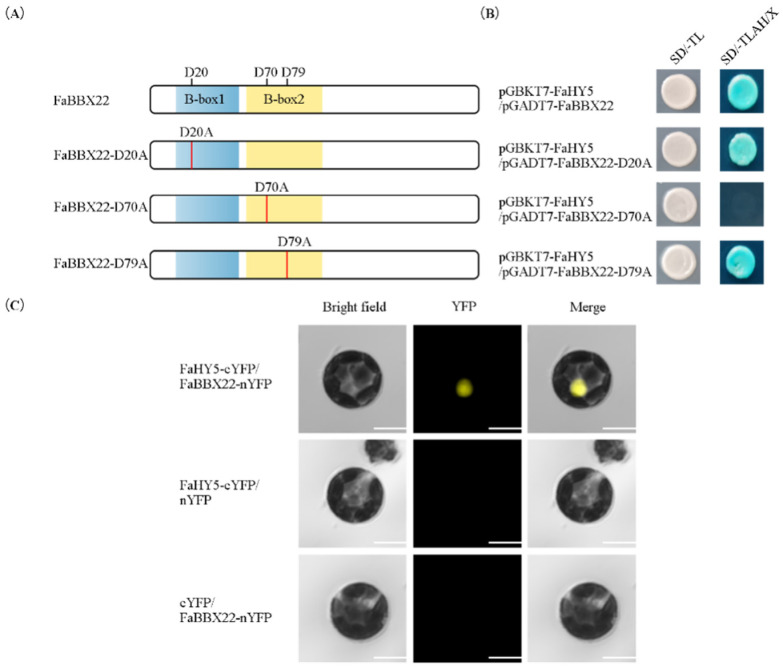
The physical interactions of FaBBX22 and FaHY5. (**A**) Schematic diagram of FaBBX22 domains and its mutations; (**B**) Yeast two-hybrid assays of FaBBX22 and FaHY5. SD/-TL indicates SD/-Trp-Leu medium, SD/-TLAH/X indicates SD/-Trp-Leu-Ade-His/X-α-gal medium with 100 ng/mL Aureobasidin A; (**C**) Bimolecular fluorescence complementation assays of FaBBX22 and FaHY5. Bars, 10 μm.

**Figure 7 ijms-23-07757-f007:**
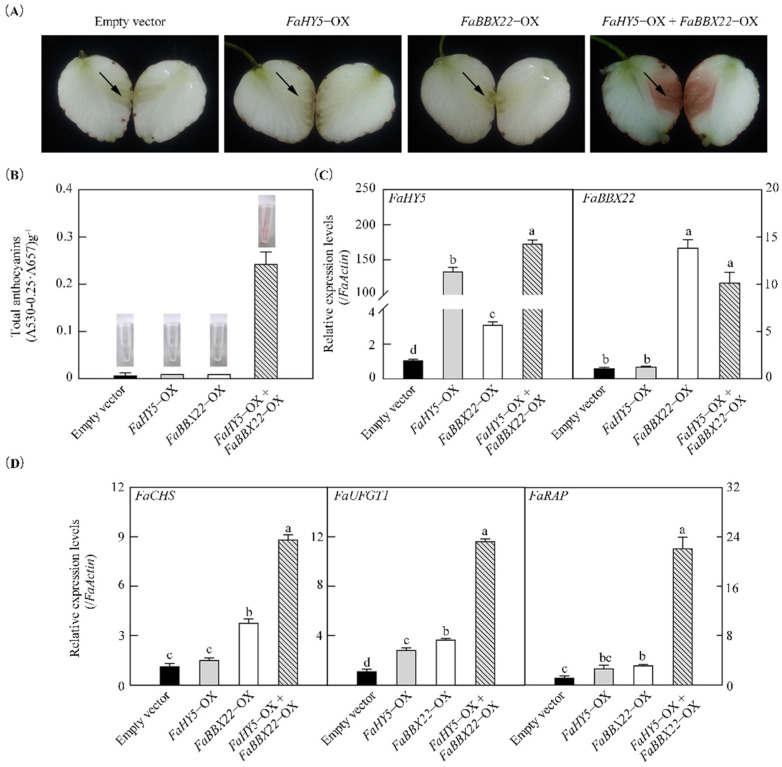
FaBBX22 and FaHY5 jointly promote the anthocyanin accumulation in white fruit strawberry variety (‘Snow white’). (**A**) Phenotypes of white flesh after infiltrating *FaBBX22*, *FaHY5* and *FaBBX22*/*FaHY5* for 5 days. The arrow points to the infiltration site; (**B**) Anthocyanin contents and extracts of strawberry fruits; (**C**) The expression levels of *FaHY5* and *FaBBX22* in different infiltration treatments; (**D**) The relative expression levels of *FaCHS*, *FaUFGT1* and *FaRAP* in different infiltration treatments. Infiltrating empty vector pCAMBIA1301−35S−Nos acts as control. Error bars represent three independent replicates. Different letters above the bars indicate significantly different values (*p* < 0.05) according to a Least Significant Difference (LSD) test.

## Data Availability

Not applicable.

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
