# Peer review of "B-Box Transcription Factor FaBBX22 Promotes Light-Induced Anthocyanin Accumulation in Strawberry (Fragaria × ananassa)"

_ijms, 2022, doi:10.3390/ijms23147757_

Round 1

Reviewer 1 Report

The manuscript describes evidence supporting the involvement of the transcription factor FaBBX22 mediating the accumulation of anthocyanin in strawberry induced by light. The manuscript is well structured, most evidence is compelling and adds to other factors reported in bibliography that activate transcription of the genes encoding enzymes of the synthesis of anthocyanins. However, there are sections of the manuscript that must be corrected and supplemented for publication.

English is generally readable, but information in some sentences must be improved. An example in the Abstract is the expression (line 15): “Here, FaBBX22, a B-box TF encoding 287 amino acids,” is misleading; one alternative could be "Here, FaBBX22 gene encoding 287 amino acids B-box TF". Another in the Abstract (line 25), “mutation of the 70th Asp” should change to “mutation of the 70th Asp to Ala”. The entire manuscript should be checked for scientific/English consistence.

Lines 159-160 and Figure 2B. Increase the size of symbols in the Figure to clearly distinguish triangles, squares, and circles. The down-up-down of the FaBBX22 level between 6 and 48 h under blue (?) light is surprising and requires explanations.

Lines 242-244. The experimental procedure for site-direct mutation must be described here or, preferably, in Materials and Methods.

Figure 7B. What bands above bars are?

Lines 407-408. Did you sequence it?  Specify.

Reviewer 2 Report

This manuscript has investigated the role of BBX22, a strawberry B-box transcription factor, in anthocyanin accumulation. The description, analysis and logical development are reasonable. Some previous reports have demonstrated that BBX TFs are involved in anthocyanin accumulation. Your result that light exposure induced anthocyanin accumulation in FaBBX22-overexpressing plants is also agriculturally important and attractive. However, the author's argument relies on results derived from overexpression assay and may not reflect the essential natural role of BBX22. Authors showed that light exposure induced anthocyanin accumulation in FaBBX22-overexpressing calli but not empty vector control (Fig. 5). In Fig. 5B, light exposure didn’t upregulate expression of FaBBX22 in the EV control, indicating that BBX22 naturally may not be involved in anthocyanin accumulation upon light exposure and so, it is possible that your result is an artifact caused by overexpression. In Fig. 1A, expression of BBX22 is higher in roots than other tissues. Taken together, it is guessed that control of anthocyanin accumulation is not an original role of BBX22. To overcome this concern, you should show any evidences which can demonstrate whether FaBBX22 naturally regulates anthocyanin accumulation. Analysis on Knockout mutant of FaBBX22 or ChIP-seq analysis to see direct regulation of the anthocyanin biosynthetic genes may be necessary.

Round 2

Reviewer 2 Report

As previously mentioned, the fact that BBX22 affects strawberry pigmentation is an important agronomic finding.

The author's responses are not necessarily complete, but it is considered to have enough content to merit publication.